# Location of Mountain Photovoltaic Power Station Based on Fuzzy Analytic Hierarchy Process—Taking Longyang District, Baoshan City, Yunnan Province as an Example

**Yiping Li [1,2], Jingchun Zhou [1,2,*] and Zhanyong Feng [1,2]**

1   Key Laboratory of Resources and Environmental Remote Sensing for Universities in Yunnan, Faculty of Geography, Yunnan Normal University, Kunming 650500, China; 15812053786@163.com (Y.L.); fengzhanyong1010@163.com (Z.F.)
2   Center for Geospatial Information Engineering and Technology of Yunnan Province, Faculty of Geography, Yunnan Normal University, Kunming 650500, China
*   Correspondence: 190005@ynnu.edu.cn

**Abstract:** Site selection is a key link in the early stage of constructing a photovoltaic power station and providing accurate guidance for the development of such stations. Taking Longyang District, Baoshan City, Yunnan Province, as an example, this article utilizes land-use status data from the third national land survey. The study focuses on five land-use types: idle land, bare land, shrub land, forest land, and another grassland, while excluding interfering land types such as construction land, ecological conservation areas, and cultivated land. Thirteen factors including terrain, weather, environment, and neighboring resources are considered. By employing the fuzzy analytic hierarchy process, a site selection model is constructed to analyze the suitability of photovoltaic power station locations. This study emphasizes the influence of geological disaster factors when selecting environmental factors. Given the high frequency of geological disasters in mountainous areas, these factors significantly affect the safety of later-stage photovoltaic power station operations. Previous research has paid less attention to this factor. The results indicate a high level of suitability for photovoltaic site selection in Longyang District, Baoshan City, with suitable, moderately suitable, and unsuitable areas accounting for 20.09%, 34.14%, and 45.77%, respectively. Previous studies have lacked sufficient validation of site selection outcomes. In this research, validation is conducted using areas where photovoltaic power stations have already been established and are under construction within the region. The accuracy of this site selection method is found to be 92.78%. The aim is to provide a scientific reference for site selection in mountainous areas with photovoltaic power station construction needs.

**Keywords:** mountain; fuzzy hierarchy; photovoltaic power station; site selection

## 1. Introduction

Many countries favor solar energy due to its convenient access and extremely low environmental pollution. China also attaches great importance to the sustainable development and utilization of solar energy. The National Energy Administration put forward policy support for photovoltaic power generation in the "Notice on Matters related to the development and construction of wind power and photovoltaic power generation in 2021" (National Energy Development New Energy (2021) No. 25) and provided a guarantee to develop and construct the project. The location of the photovoltaic power station is a critical step in the power station's construction. An improper location will reduce the power station's power generation and operating life, increase investment, operation, and maintenance costs, and will also cause adverse effects on the surrounding environment. Therefore, exploring the location of photovoltaic power plants and their precautions is crucial to improve the energy efficiency and sustainable utilization of photovoltaic power plants.

Yunnan Province is located on the southwestern border of China, with highland and mountainous areas accounting for over 95% of its total land area. The average elevation is around 2000 m. Within its jurisdiction, most of the 16 prefecture-level cities are typical highland and mountainous cities [1]. In recent years, with the growth in population, Yunnan Province has seen a significant increase in its electricity demand. However, fossil fuels such as coal and oil still make up 52.2% of the total energy consumption. To drive a shift in development patterns and accelerate the utilization of green energy sources, solar power has become crucial [2]. As one of the regions abundant in solar energy, photovoltaic power stations have become the preferred choice, and selecting the most suitable locations for solar power plants is of utmost importance.

The traditional site selection method of the power station is mainly based on field investigation and local guidance, and land selection will be biased due to the professional ability of researchers. In order to eliminate the limitations of traditional methods, location model construction has attracted much attention from scholars at home and abroad in recent years. From the perspective of research methods, the main methods include the Analytic Hierarchy Process (AHP) and multi-criteria decision making (MCDM). Sun et al. calculated factor weights to evaluate the land suitability of photovoltaic power stations by considering protected areas, topography, and water resources [3]. Solangi et al. employed the AHP to determine the weights of economic, environmental, location, climate, and topographical factors involved in decision making. They evaluated the location of solar photovoltaic power generation projects in Pakistan [4]. Ma utilized the AHP method to study photovoltaic siting in Yalong River Basin and analyzed the most suitable area for building stations [5]. Doljak et al. determined weights based on terrain, vegetation, climate, and humidity using AHP and combined it with MCDM to determine the best construction site for photovoltaic power stations [6]. Lurwan et al. adopted a multi-criterion decision-making method to determine the best location for large-scale grid-connected photovoltaic power stations based on slope, road, power grid, solar radiation, and other conditions [7]. Rediske et al. adopted MCDM and AHP to weigh the factors such as water resources, roads, slopes, and protected areas to evaluate the suitability of large photovoltaic power plants in Brazil. They concluded that the most suitable station building area was near the substation with a low slope [8]. Majumdar et al. employed MCDM to evaluate the suitability of photovoltaic development land in Arizona based on terrain, location, and solar energy resources [9]. Elboshy et al. considered the constraints of topography, environment, meteorology, and climate when evaluating the suitability of different regions in Egypt for the construction of photovoltaic power stations. Among the 10 criteria selected, they particularly emphasized the impact of lightning flash rate on the selection of photovoltaic sites in Egypt [10]. However, in mountainous areas, geological disasters are unavoidable, and the displacement of photovoltaic arrays due to geological disasters can disrupt normal operations [11]. This factor is crucial in the site selection process for photovoltaic power stations, but it was not taken into account in the aforementioned study as a key criterion.

The selection of land for photovoltaic purposes varies due to different local policies and actual geographical conditions. In existing studies, land-use types have been included as factors and incorporated into decision-making models [12]. Some research has even substituted land-use types with land-use costs. Based on local land policies and practical requirements, agricultural land, urban areas, and ecological regions were excluded from consideration during the site selection process, with focus placed on other land-use types [13]. In this study, in accordance with Yunnan Province's land-use policies and based on the Third National Land Use Survey, five land-use types were chosen, namely vacant land, bare land, shrubland, other forestland, and other grassland. This approach avoided constraints related to arable land and ecological regions, reducing land-use costs and ensuring the rational utilization of land without impacting existing land-use patterns.

The AHP method divides the problem into decision and goal levels according to the nature of the problem and the overall goal to construct a multi-level analysis structure model and order the problems relative to the importance weight or relative advantages and

disadvantages of the target level [14]. MCDM is a systematic method that takes the relevant influencing factors of the decisive goal as the decision criteria for comprehensive analysis and finally comes up with the decision scheme. After unifying the dimensions, the method integrates and analyzes all kinds of criteria from a macroscopic perspective [15]. Applying AHP and MCDM to verify the suitability of photovoltaic power station siting is more scientific than traditional siting methods. However, it is more subjective in determining the weight of decision factors and evaluating the consistency of the siting model is difficult [16]. The fuzzy Analytic Hierarchy Process overcomes the main shortcoming of AHP, which is that the matrix consistency can be judged only after repeated data adjustment and multiple tests. The index score deviation caused by human preference in MCDM is reduced, and the analysis results become more scientific and reasonable. This method is highly accurate and applicable in the selection process of Antarctic research stations [17], At the same time, it has also been accurately verified in the selection of wind power generation sites in Sudan [18].

This paper employs the fuzzy Analytic Hierarchy Process (FAHP) and GIS Spatial analysis to study the site selection model of photovoltaic power stations in Longyang District, Baoshan City, Yunnan Province, in complex mountainous areas to explore suitable areas for the site selection of photovoltaic power stations and provide theoretical basis and practical reference for the site selection of similar projects.

## 2. Materials and Methods

### 2.1. Study Area

Longyang District, Baoshan City, Yunnan Province, shown in Figure 1, is located in the southern part of the Qinghai–Tibet Plateau at the tail of the Nujiang Mountains and among the Gaoligong Mountains. The mountains and roads in the territory are undulating, which is a typical subtropical plateau climate type. Coupled with the terrain of low latitude, high altitude, and significant elevation difference, Longyang District forms a three-dimensional climate of "one mountain divided into four seasons and ten li different days". It has a low loss of solar radiation in the atmosphere, which is conducive to the efficient use of solar energy and stable output of electric energy. It is one of the most suitable areas for developing the photovoltaic energy industry in Yunnan Province. In recent years, Longyang District has vigorously promoted the construction of photovoltaic projects and has constructed two photovoltaic power stations with a total installed capacity of 80,000 kilowatts. In 2022, we started the construction of five photovoltaic power generation projects: Pupiao, Hengshan, Luoming, Yangliu, and United. Regarding the large-scale development of the photovoltaic industry, Longyang District can also arrange more photovoltaic power stations to meet the power demand for residents and enterprises and drive the development of mountain areas. Therefore, the location planning of photovoltaic power stations is crucial for Baoshan City to alleviate the power industry's environmental pressure, supplement the surrounding areas' energy supply, and promote regional economic development.

### 2.2. Data Preparation

The data employed in this study mainly include DEM, slope, slope direction data, meteorological, disaster, hydrological data, administrative division, residential area, road data, land use, and ecological protection area data. All the data were processed and converted into raster data with a spatial resolution of 30 m × 30 m, as shown in Table 1.

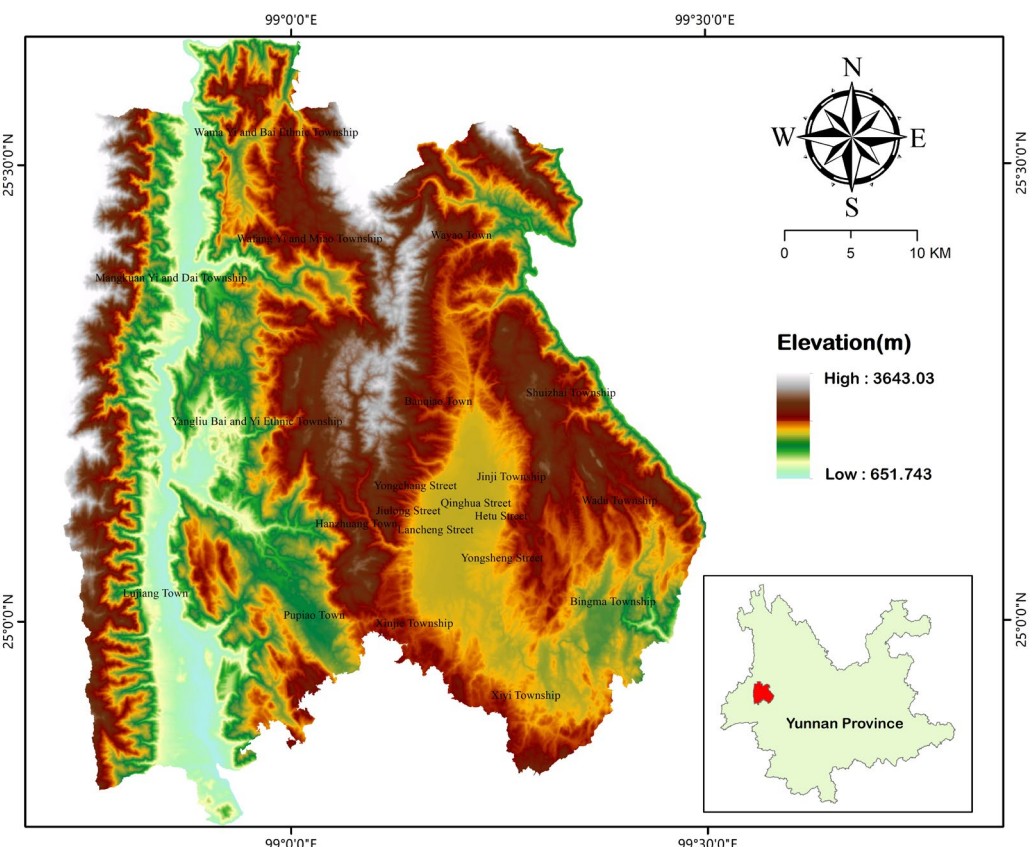

**Figure 1.** Location map of Longyang District, Baoshan City.

**Table 1.** Basic data of the research area.

| Data Name | Data Source |
| --- | --- |
| DEM date | Geospatial data cloud platform (http://www.gscloud.cn/, 20 May 2023) |
| Slope, slope direction | Generated by DEM data |
| Administrative division data | DIVA-GIS website (http://data.diva-gis.org/, 22 May 2023) |
| Meteorological, disaster, and hydrological data | Resource Environmental Science and Data Center (https://www.resdc.cn/, 21 May 2023) |
| Residential data, road data | National geographic information resources directory service system (https://www.webmap.cn/, 23 May 2023) |
| Land use data, ecological protection area data | Data of the third National Land Survey |

### 2.3. Construction of PV Location Constraints and Evaluation Index System

#### 2.3.1. Constraints of Land Use Type

Land use type is an essential factor influencing the location of solar power plants [19]. According to the existing policies for the protection of cultivated land, the site selection of the existing photovoltaic power station will avoid cultivated land and try to construct the non-cultivated land such as desert and Gobi wasteland with abundant sunlight [20], which should not compete with agricultural land, housing construction land, and water bodies. The construction of a photovoltaic power station should also avoid geological disaster areas. A specific distance should be kept from residential areas and roads to ensure that traffic is not destroyed by the surrounding landscape pattern [21]. In order to minimize the disturbance of photovoltaic panels on the normal life of residents, the installation site should be a certain distance from residential areas. Simultaneously, to

reduce the transportation costs of electricity, the distance from residential areas should not be too far, so a distance within 5 km is most suitable [22]. To ensure the preservation of the surrounding landscape and isolation from roads, the selection of the photovoltaic power station site should maintain a certain distance from roads. Additionally, to facilitate the installation and transportation of equipment, this distance should not be excessively far [23]. In areas where land is in short supply, to construct the photovoltaic power stations, part of agricultural land can be selected to realize the efficient use of land by photovoltaic agricultural mode. However, this technology should consider local topography, land-use policy, cost input of photovoltaic enterprises, and sustainable agricultural production in the later period. In addition, relevant agricultural industry standards should be improved when selecting land types [24].

### 2.3.2. Topography and Geomorphology Constraints

The shape of the land largely determines the difficulty of installing photovoltaic modules, which in turn limits the total installed capacity and is also the main factor considered by the photovoltaic power station site. In the previous literature, the higher the altitude, the more challenging the installation of solar photovoltaic panels becomes [25]. Aly argued that the ground slope of photovoltaic power stations should not exceed 2.1%. For large-scale photovoltaic power stations, an acceptable slope percentage falls within the range of 3% to 11%. In the case of mountainous photovoltaic construction, slopes with actual inclinations less than 25% are considered feasible. Photovoltaic power stations should be selected in open areas, and the massif orientation determines the quality of the receiving light conditions. In countries in the northern hemisphere, the optimal direction for placing solar photovoltaic panels is generally south or east. Therefore, the construction of photovoltaic power stations should choose the south and east directions without considering the northern areas [26].

### 2.3.3. Environmental Constraints

The construction of a photovoltaic power station will harm the soil and water resource ecology of the station and surrounding areas [27]. Therefore, the photovoltaic project must ensure that vegetation under the battery module array is kept unchanged while meeting its regular operation and avoiding surface exposure, hardening, or other uses. The station building area should have a specific distance from the existing water area and avoid the environmental protection area. At the same time, to facilitate power transportation in the later stage and be connected to the grid, the proximity of the PV power plant to the substation will help reduce the cost of power transportation. The distance between the site and the nearby substation should also be employed as a constraint condition [28].

### 2.3.4. Constraints on Natural Conditions

Solar radiation is the most popular factor affecting the power generation potential. Economic benefits of photovoltaic power stations should choose the area with abundant solar radiation for station construction. Under sufficient lighting conditions, the air temperature level will restrict the normal operation of photovoltaic modules [29], and the soaking of photovoltaic materials by rain on rainy days will also affect the productivity of photovoltaic modules [30]. When the photovoltaic station is built, sufficient ambient temperature is conducive to the power generation of photovoltaic panels, while excessive precipitation leads to a reduction in air temperature, which is not conducive to the normal operation of photovoltaic power plants. Therefore, temperature and precipitation should be considered when constructing photovoltaic stations [31].

### *2.4. The Suitability Assessment Index System for Photovoltaic Site Selection*

In summary, based on foreign scholars' consideration of various factors in photovoltaic site selection [32,33], 13 influencing factors such as land-use type, elevation, slope, slope direction, hydrology, geological hazards, temperature, and sunshine hours were selected to

participate in the site selection decision. The land-use type avoids the land-use constraint area. It selects five land types, including idle, bare, shrub, other forest, and grasslands, to establish a bipolar evaluation index system for photovoltaic site selection. The first-level index includes four categories of topography, meteorological conditions, environmental conditions, and adjacent resources, and the second-level index is divided according to nature or type, as shown in Table 2.

**Table 2.** Evaluation index system.

| Goal | Primary Index Layer | Secondary Index Layer |
|---|---|---|
| | Landform | Dem |
| | | Slope |
| | | Slope orientation |
| | Meteorological condition | Annual average surface temperature |
| | | Annual average temperature |
| | | Annual precipitation |
| Evaluation of PV location suitability in Longyang District, Baoshan City | | Annual average sunshine duration |
| | Environmental condition | Distance from water |
| | | Distance from ecological protection zone |
| | | Distance from geological hazards |
| | Adjacent resources | Distance to residential areas |
| | | Distance from highway |
| | | Distance from substation |

Each assessment factor was treated differently based on the complex landform in Longyang District of Baoshan City. The classification assignment of each secondary indicator was determined based on the actual suitability of each influencing factor. The appropriate assignment was 4, the generally appropriate assignment was 3, the inappropriate assignment was 2, and the very inappropriate assignment was 1. Table 3 shows the specific suitability level.

**Table 3.** Suitability level of photovoltaic site selection indicators in Longyang District.

| Index | Suitable (4) | Generally Suitable (3) | Unsuitable (2) | Very Unsuitable (1) |
|---|---|---|---|---|
| Dem | ≤1000 m | 1001–1800 m | 1801–3000 m | >3000 m |
| Slope | 0°–10° | 10°–15° | 15°–25° | >25° |
| Slope orientation | South, east, southeast | Southwest and northeast | West and northwest | North |
| Annual precipitation | 686–858 mm | 859–946 mm | 946–1230 mm | |
| Annual average temperature | 16–20 °C | 8–15 °C | >20 °C | |
| Annual average surface temperature | 17–19 °C | 15–17 °C | 12–15 °C | |
| Annual average sunshine duration | 2201 h–2400 h | 2001 h–2200 h | <2000 h | |
| Distance from water | >300 m | | | ≤300 m |
| Distance from ecological protection zone | >500 m | | | ≤500 m |
| Distance from geological hazards | >1000 m | 501–1000 m | 300–500 m | <300 m |
| Distance to residential areas | 1501 m–5000 m | 801–1500 m | 501–800 m | 0–500 m, >5000 m |
| Distance from highway | 1501 m–5000 m | 801–1500 m | 501–800 m | 0–500 m, >5000 m |
| Distance from substation | 0–1 km | 1–3 km | >3 km | |

### 2.5. Materials and Methods

The data flow chart of this study is shown in Figure 2.

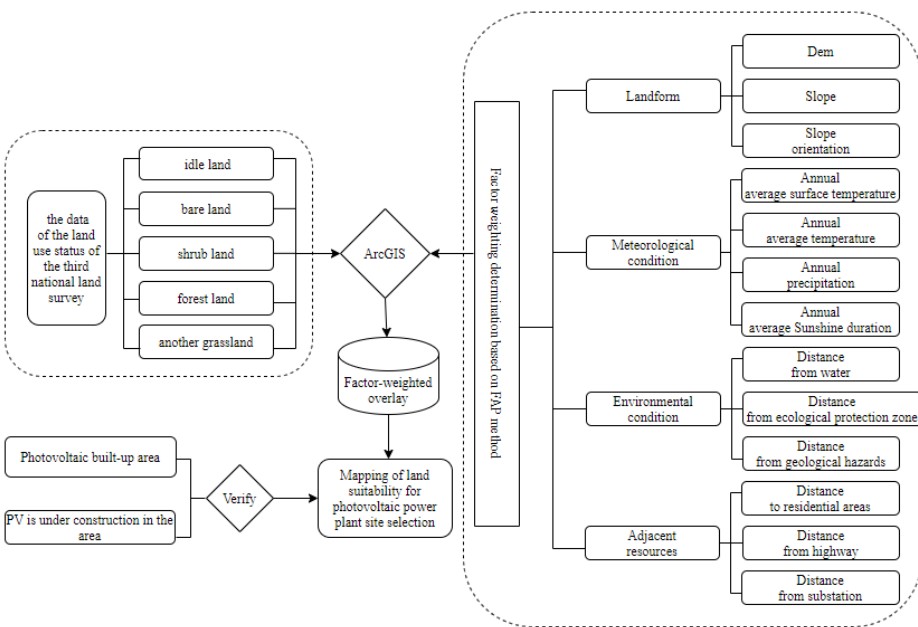

**Figure 2.** Longyang District photovoltaic power station site selection evaluation process.

### 2.5.1. Fuzzy Analytic Hierarchy Process

The fuzzy Analytic Hierarchy Process (F-AHP) introduces the concept of a fuzzy set in the mathematical model. It transforms the relevant indicators involved in decision making from qualitative to quantitative analysis. It overcomes the shortcomings of the Analytic Hierarchy Process (AHP), which requires repeated adjustments of data and multiple tests to determine matrix consistency. The fuzzy AHP significantly reduces the deviation in indicator scores caused by human preferences, making the analytical results more scientifically and logically sound. The constructed fuzzy consistent judgment matrix R compares the relative importance of a particular element in the previous criterion layer with its related elements at this level. Suppose element A of the previous level and element b1 of this level. When comparing two elements in the bn correlation and fuzzy consistent judgment matrix, the fuzzy relationship is described with a membership degree of "element A is much more important than element c". This study employs a scale of 0.1–0.9 to judge the fuzzy relationship, which is explained as follows in Table 4.

**Table 4.** Quantity Scale.

| Scale | Definition | Implication |
|-------|-----------|-------------|
| 0.5 | Equally important | Both elements are equally important when compared |
| 0.6 | Slightly important | Comparing two elements, one element is slightly more important than the other |
| 0.7 | Obvious importance | One element is significantly more important than the other |
| 0.8 | More important | Comparing the two elements, one element is much more important than the other |
| 0.9 | Extremely important | Comparing two elements, one element is extremely important than the other |
| 0.1, 0.2, 0.3, 0.4 | Inverse comparison | If the judgment $r_{ij}$ is obtained by comparing element $a_i$ with element $a_j$, then the judgment obtained by comparing element $a_j$ with element $a_i$ is $r_{ij} = 1 - r_{ij}$ |

(1)  The fuzzy complementary matrix is obtained as follows:

$$
R = \begin{bmatrix}
r_{11} & \cdots & r_{1j} & \cdots & r_{1n} \\
\vdots & & \vdots & & \vdots \\
r_{i1} & \cdots & r_{ij} & \cdots & r_{in} \\
\vdots & & \vdots & & \vdots \\
r_{n1} & \cdots & r_{n_j} & \cdots & r_{nn}
\end{bmatrix}
\tag{1}
$$

The R matrix satisfies: $r_{ij}$ is a scale value of 0.1–0.9, $i = 1,2,\cdots,n$, $j = 1,2,\cdots,n$.

Additionally, $r_{j_i} + r_{ij} = 1$; When $i = j$, $r_{ij} = 0.5$; $r_{ij}$ indicates the membership degree that element ai is more important than element aj, and a larger $r_{ij}$ indicates that element ai is more important than element aj.

(2)  The value $w_i$ of the index weight vector W in the first-level criterion layer is obtained by solving the fuzzy matrix R through row normalization. Consider n first-level criterion layer factors. Then, the index weight W is expressed as follows:

$$
W = w_1 \; w_2 \; w_3 \cdots w_i \cdots w_n
$$

$$
w_i = \frac{1}{n} - \frac{1}{2\alpha} + \frac{1}{n\alpha}\sum_{k=1}^{n} r_{ik}
\tag{2}
$$

where n is the order of the fuzzy matrix, and $\alpha$ is the measure of the difference ($w_{a_i} - w_{a_j}$) between the importance of elements ai and aj. The larger the (Wi, Wj), the less attention decision makers pay to the importance differences between indicator layers. In practice, $\alpha$ is $(n^{-1})/2$. Similarly, the weight method in the second-level index layer is compatible with the solution in the first-level criterion layer.

(3)  Various verification methods have been proposed for fuzzy matrix consistency judgment. This paper performs the matrix consistency test according to Theorems of sufficient and necessary conditions for the fuzzy matrix consistency judgment proposed in the literature [34]. This test method is more accurate, scientific, and simple than obtaining the maximum eigenvalue of a matrix and its corresponding eigenvector.

(4)  Determine the weight vector

When the criterion layer fuzzy matrix R and the secondary index matrix Ri meet the consistency requirements, the consistency of the comprehensive evaluation matrix also meets the requirements. The final weights of factors in the second-level index layer can be expressed as follows:

$$
w_{Ai} = \sum_{i=1}^{n} w_i w_{ip}
\tag{3}
$$

where $w_{Ai}$ represents the final weight of each factor in the second-level indicator layer, $w_i$ represents the weight of the first-level criterion layer to which the indicator belongs, and $w_{ip}$ represents the membership weight of each factor in the second-level indicator layer.

2.5.2. Determining the Weight Factor Based on the Fuzzy Analytic Hierarchy Process

The project leader of this research invited experts with backgrounds in engineering, environmental science, and sociology to form an expert advisory committee. The committee assessed the importance of 13 factors related to photovoltaic site selection, and the levels of significance for each factor are presented in Table 5. Based on the importance levels indicated in Table 5, corresponding to the quantitative scale in Table 4, fuzzy judgment matrices for each level of indicators were constructed as follows.

**Table 5.** Table of importance of evaluation indicators.

| Evaluation Indicators | Landform | Meteorological Condition | Environmental Condition | Adjacent Resources |
|---|---|---|---|---|
| Landform | Equally important | Obvious importance | Inverse comparison | Slightly important |
| Meteorological condition | Inverse comparison | Equally important | Inverse comparison | Inverse comparison |
| Environmental condition | Slightly important | More important | Equally important | Obvious importance |
| Adjacent resources | Inverse comparison | Slightly important | Inverse comparison | Equally important |
| **Evaluation indicators** | **Dem** | **Slope** | **Slope orientation** | |
| Dem | Equally important | Obvious importance | Inverse comparison | |
| Slope | Inverse comparison | Equally important | Inverse comparison | |
| Slope orientation | Slightly important | More important | Equally important | |
| **Evaluation indicators** | **Annual average surface temperature** | **Annual average temperature** | **Annual precipitation** | **Annual average Sunshine duration** |
| Annual average surface temperature | Equally important | Inverse comparison | Inverse comparison | Inverse comparison |
| Annual average temperature | More important | Equally important | Slightly important | Obvious importance |
| Annual precipitation | Obvious importance | Inverse comparison | Equally important | Slightly important |
| Annual average Sunshine duration | Slightly important | Inverse comparison | Inverse comparison | Equally important |
| **Evaluation indicators** | **Distance from water** | **Distance from ecological protection zone** | **Distance from geological hazards** | |
| Distance from water | Equally important | Slightly important | More important | |
| Distance from ecological protection zone | Inverse comparison | Equally important | Obvious importance | |
| Distance from geological hazards | Inverse comparison | Inverse comparison | Equally important | |
| **Evaluation indicators** | **Distance to residential areas** | **Distance from highway** | **Distance from substation** | |
| Distance to residential areas | Equally important | Slightly important | Inverse comparison | |
| Distance from highway | Inverse comparison | Equally important | Inverse comparison | |
| Distance from substation | Slightly important | Obvious importance | Equally important | |

According to the calculation method of factor weights in the fuzzy analytic hierarchy process, the fuzzy judgment matrix of all indicator levels is constructed as follows:

$$R = \begin{bmatrix} 0.5 & 0.7 & 0.4 & 0.6 \\ 0.3 & 0.5 & 0.2 & 0.4 \\ 0.6 & 0.8 & 0.5 & 0.7 \\ 0.4 & 0.6 & 0.3 & 0.5 \end{bmatrix} \quad R_1 = \begin{bmatrix} 0.5 & 0.7 & 0.4 \\ 0.3 & 0.5 & 0.2 \\ 0.6 & 0.8 & 0.5 \end{bmatrix}$$

$$R_2 = \begin{bmatrix} 0.5 & 0.2 & 0.3 & 0.4 \\ 0.8 & 0.5 & 0.6 & 0.7 \\ 0.7 & 0.4 & 0.5 & 0.6 \\ 0.6 & 0.3 & 0.4 & 0.5 \end{bmatrix} \quad R_3 = \begin{bmatrix} 0.5 & 0.6 & 0.8 \\ 0.4 & 0.5 & 0.7 \\ 0.2 & 0.3 & 0.5 \end{bmatrix} \quad R_4 = \begin{bmatrix} 0.5 & 0.6 & 0.4 \\ 0.4 & 0.5 & 0.3 \\ 0.6 & 0.7 & 0.5 \end{bmatrix}$$

According to the theorem in the fuzzy matrix consistency test, the fuzzy matrix of the criterion layer and index layer meets the experimental consistency requirements, corresponding to Formulas (1)–(3) of the factor weights in the fuzzy analytic hierarchy

process, and the normalization process is then performed to obtain the index values of each level, as shown in Table 6:

**Table 6.** Evaluation index system and weights.

| Goal | Primary Index | First-Order Index Weight | Secondary Index | Secondary Index Weight | Comprehensive Weight |
|---|---|---|---|---|---|
| Evaluation of PV location suitability in Longyang District, Baoshan City | Landform | 0.283 | Elevation | 0.367 | 0.104 |
| | | | Slope | 0.167 | 0.047 |
| | | | Slope orientation | 0.467 | 0.132 |
| | Meteorological condition | 0.150 | Annual average Surface temperature | 0.150 | 0.023 |
| | | | Annual average temperature | 0.350 | 0.053 |
| | | | Annual precipitation | 0.283 | 0.043 |
| | | | Annual average Sunshine duration | 0.217 | 0.033 |
| | Environmental condition | 0.350 | Distance from water | 0.167 | 0.058 |
| | | | Distance from ecological protection zone | 0.366 | 0.128 |
| | | | Distance from geological hazards | 0.467 | 0.163 |
| | Adjacent resources | 0.217 | Distance to residential areas | 0.333 | 0.072 |
| | | | Distance from highway | 0.233 | 0.051 |
| | | | Distance from substation | 0.433 | 0.094 |

## 3. Result Analysis and Verification

### 3.1. Data Processing Results

In this study, all data were resampled to a spatial resolution of 30 m × 30 m as raster data. In ArcGIS 10.8, the selected 13 factors were processed using buffer analysis based on the criteria defined in Table 3. The 13 factor raster data were clipped using the five land-use types selected, as shown in Figure 3. After determining the suitability levels for factor subdivisions, an overlay analysis was conducted using the weights assigned to each factor through the fuzzy Analytic Hierarchy Process (FAHP) in the "Spatial Analyst Tools—Map Algebra—Raster Calculator" of ArcGIS. The results are depicted in Figure 4.

### 3.2. Result Analysis

Based on the land-use status data of the third National Land Survey, this study avoids the restricted land types such as urban construction areas, ecological reserves, and cultivated land. ArcGIS software was employed to screen out the empty land, bare land, shrubland, other grassland, and other forests, and a weighted superposition analysis was performed with the four first-level indicators and thirteen s-level sub-criteria according to the factor weights determined in Table 5 to determine the suitability of the site selection region. On the premise of the unchanged nature of the original land use, based on the principles of saving and intensive land use, ensuring the scale of photovoltaic station construction and the sustainable use of the original vegetation, the suitability analysis was performed on the map spots with an area larger than 1 hectare. The natural breakpoint method was adopted to grade the results, and the suitability of the location of photovoltaic power stations in Longyang District was obtained, as shown in Figure 3 and Table 7.

**Table 7.** Suitability analysis of photovoltaic site delection in Longyang District.

| Location Suitability Analysis | Suitable | Generally Suitable | Unsuitable |
|---|---|---|---|
| Number of pixels | 35,201 | 59,782 | 80,179 |
| Area (ha) | 3168.09 | 5380.38 | 7216.11 |
| Proportion (%) | 20.09 | 34.14 | 45.77 |

According to the suitability analysis results, Figure 5 is obtained in units of townships. The suitable areas for PV station construction are concentrated in Yangliubai Yi Township, Pupiao Township, Lujiang Township, and Mengkuan Yi Dai Township in the southwest of Longyang District, accounting for 25.3%, 16.1%, 8.4%, and 7.8% of the total suitable area, respectively. The generally suitable areas accounted for 29.8%, 15.5%, 12.1%, and 10.1% of the total generally suitable areas.

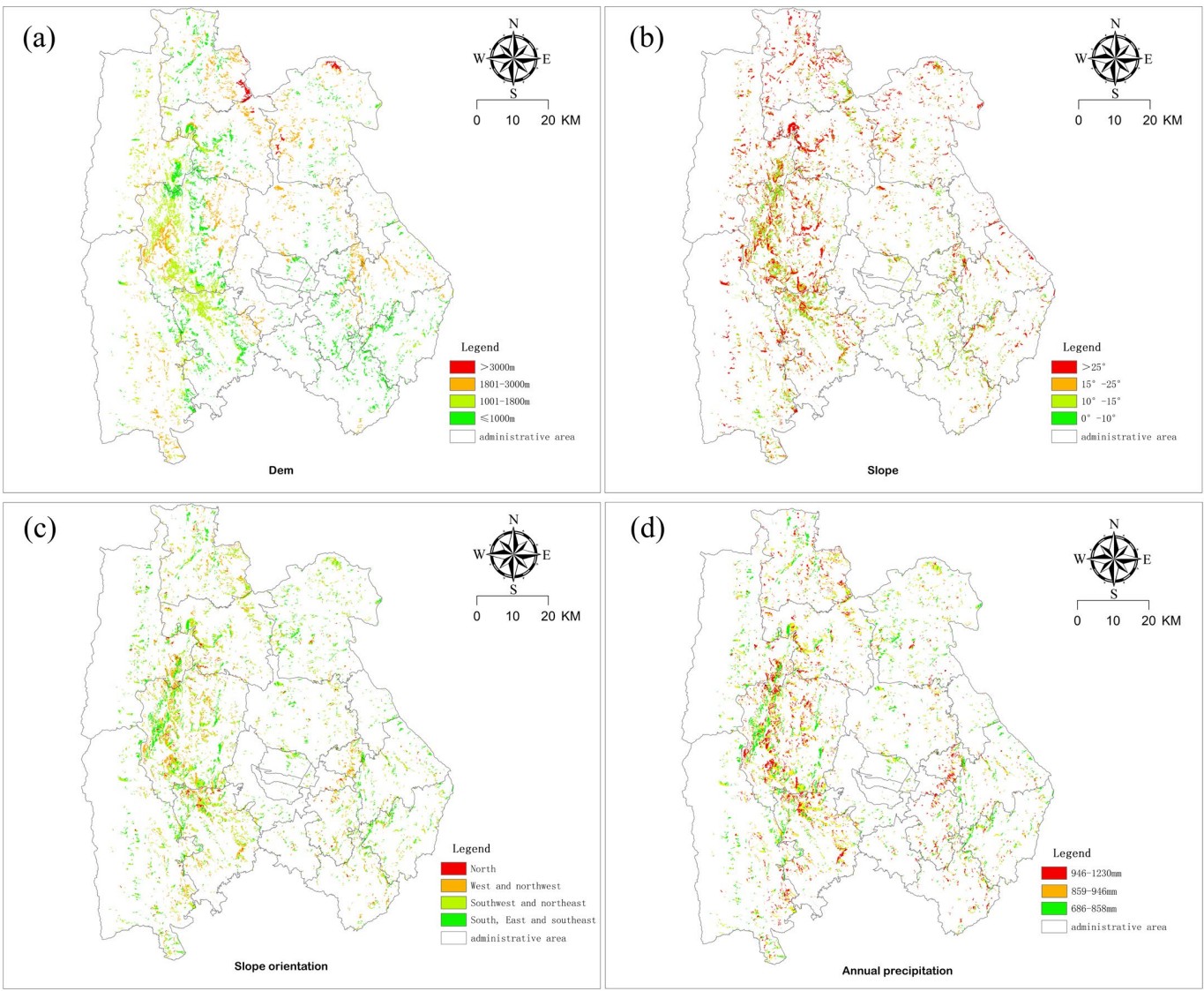

**Figure 3.** *Cont.*

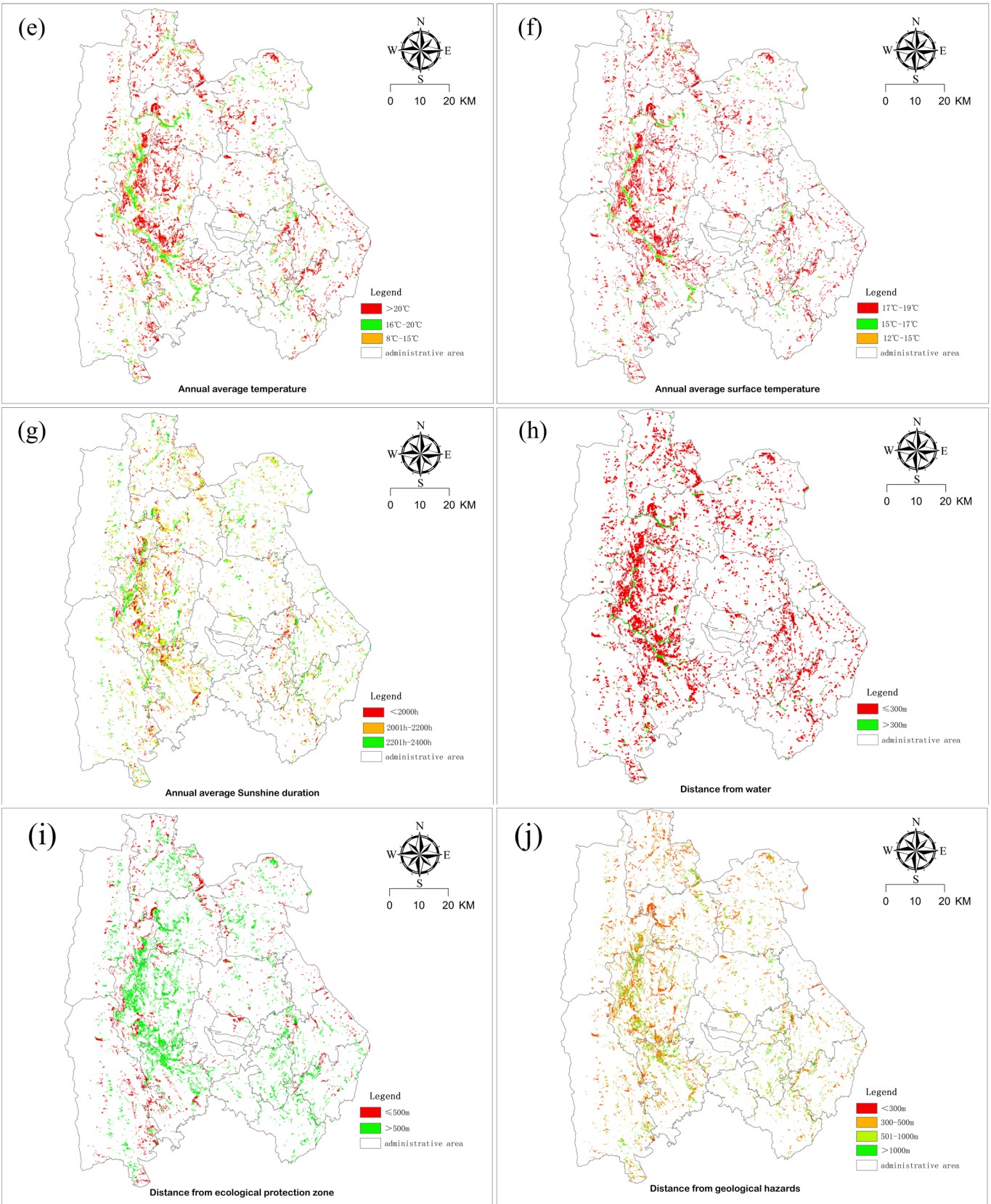

**Figure 3.** *Cont.*

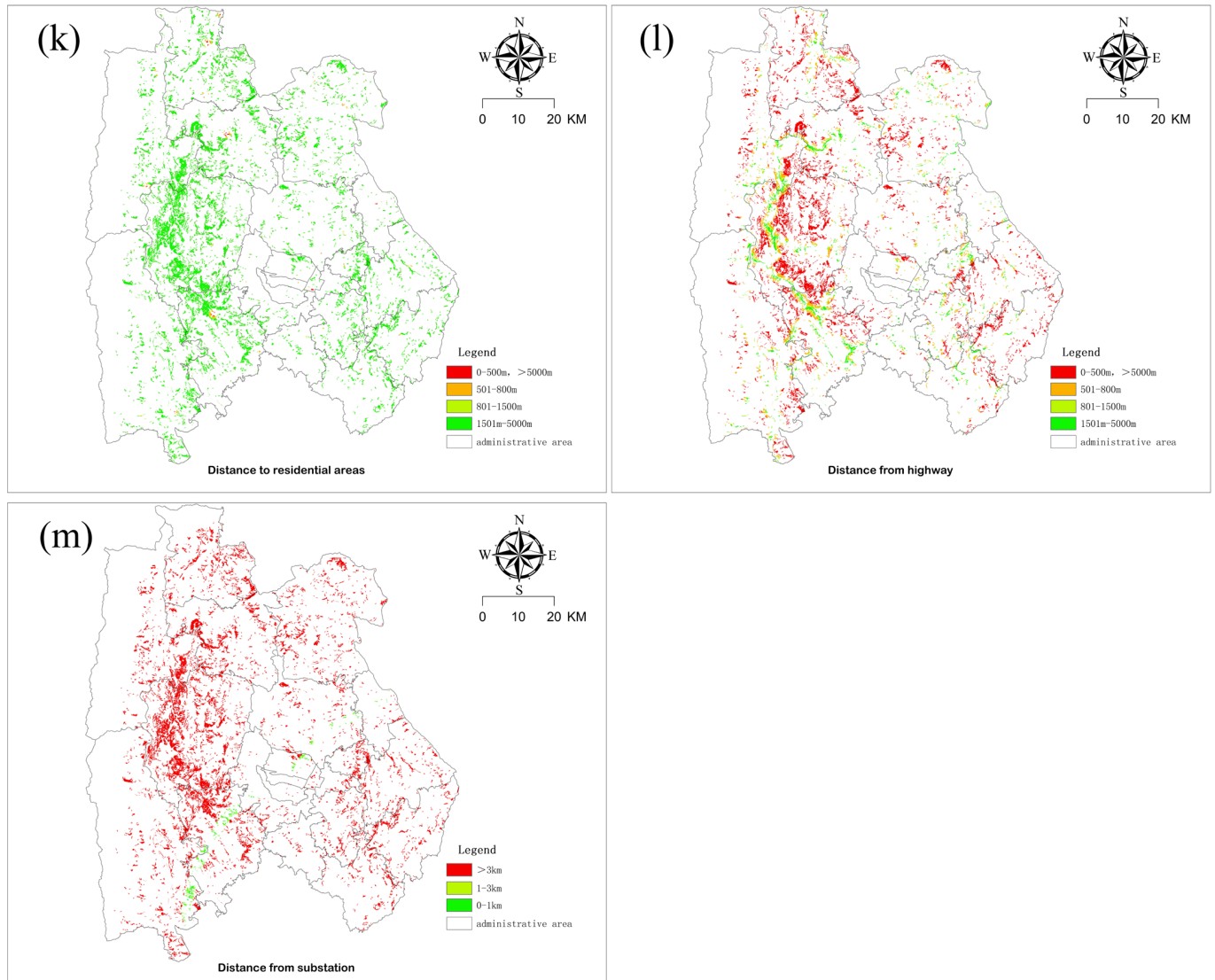

**Figure 3.** Processing chart of 13 evaluation factors in fuzzy Analytic Hierarchy Process. Input datasets: (**a**) Dem of longyang, (**b**) Slope derived from Dem, (**c**) Slope orientation from Dem, (**d**) Annual precipitation, (**e**) Annual average temperature, (**f**) Annual average Surface temperature, (**g**) Annual average Sunshine duration, (**h**) Distance from water, (**i**) Distance from ecological protection zone, (**j**) Distance from geological hazards, (**k**) Distance to residential areas, (**l**) Distance from highway, (**m**) Distance from substation.

Photovoltaic power plants are usually constructed in suitable and generally suitable areas. This area has a moderate elevation and a slope between 5° and 10°. The plateau differs from the plain area, and the slopes to the southeast promote the full use of solar energy. The annual precipitation of about 650 mm in the region reduces the damage to photovoltaic materials caused by rain soaking, and the average annual temperature of 18 °C ensures the normal production efficiency of photovoltaic modules. The suitable construction area is about 500 m away from the water and the occurrence point of natural disasters, significantly ensuring that the water ecology is not destroyed and reducing the loss caused by sudden natural disasters in the later period. At the same time, there is a moderate distance from residential areas and roads, which ensures that residential life is not disturbed and the surrounding road environment is not polluted.

**Figure 4.** Suitability level analysis of photovoltaic site selection in Longyang District, Baoshan City.

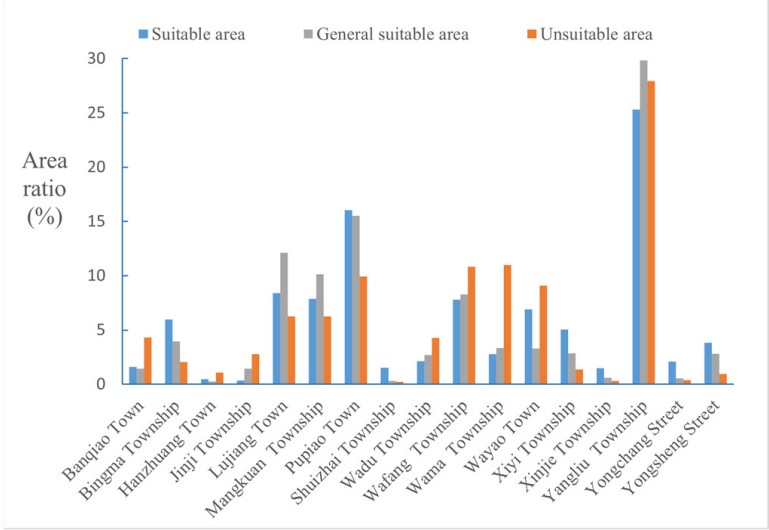

**Figure 5.** Analysis of the proportion of suitable areas in each township.

According to the photovoltaic site selection suitability analysis of five land types, including idle land, bare land, shrubland, other forest lands, and other grassland, Figure 6 shows that shrubland occupies the maximum proportion, accounting for 69% of the total selected area, and is located in the southwest of Longyang District, followed by other grassland, accounting for 15.8%. The minimum proportion of bare land is only 0.3%. The suitable probability of idle land is large, and this kind of land has significant advantages as the land for photovoltaic station construction, which saves costs for enterprises, rationally develops and utilizes the land, and performs vegetation restoration on the ground in the later stage, reducing the land exposure and promoting environmental improvement. The unsuitable area of shrubland accounted for 41.39%, about 4445.1 hectares, mainly because the shrubland in this region was close to the water protection area and less than 500 m away from the environmental protection area, resulting in the unsuitability of most of the areas for construction. Other unsuitable grassland areas accounted for 22.18% of the total grassland area, about 1494.21 hectares. Most of this area is located on steep slopes, and it is challenging to construct photovoltaic power stations with slopes greater than 25°, so site selection of the land is unsuitable.

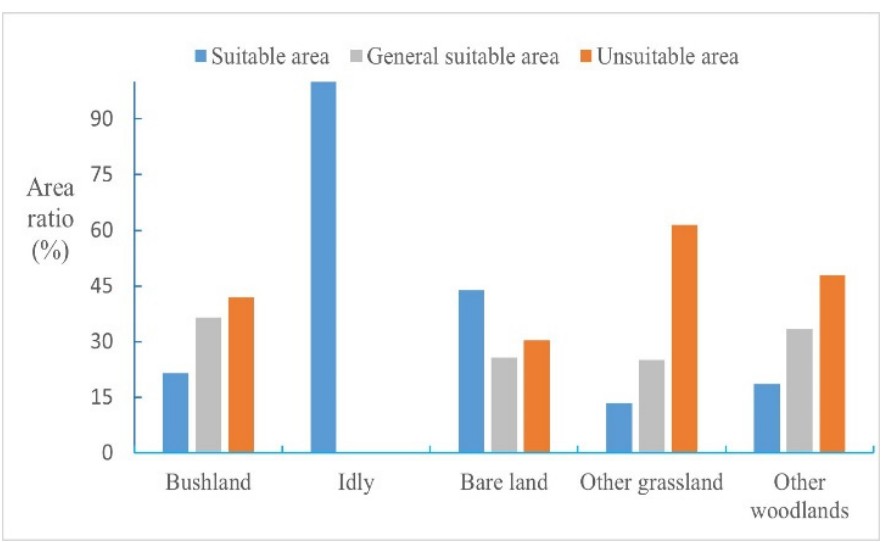

**Figure 6.** Suitability analysis of various land types.

### *3.3. Verification of the Suitability of Longyang District Photovoltaic Site Selection*

The location of constructed or under-construction photovoltaic power stations in the research area is utilized to verify the research results, as shown in Figure 7.

The accuracy of photovoltaic site selection significantly impacts the success and benefits of solar power generation projects. Precise site selection ensures that solar panels are adequately exposed to sunlight, maximizing the absorption of solar energy. This, in turn, enhances energy output, increasing the efficiency of the power generation system [35]. The return on investment for photovoltaic projects is directly linked to energy production, and accurate site selection guarantees optimal system performance, leading to higher electricity generation and increased ROI [36]. Simultaneously, accurate site selection not only helps mitigate adverse impacts on the natural environment by avoiding construction in ecologically vulnerable areas, reducing ecosystem disruption, but also enhances project sustainability [37]. It minimizes the costs, time, and resources required for the construction of solar photovoltaic fields to the greatest extent possible [38]. The results of this study are verified by using the locations of the completed and under construction photovoltaic power stations in the study area, as shown in Figure 7.

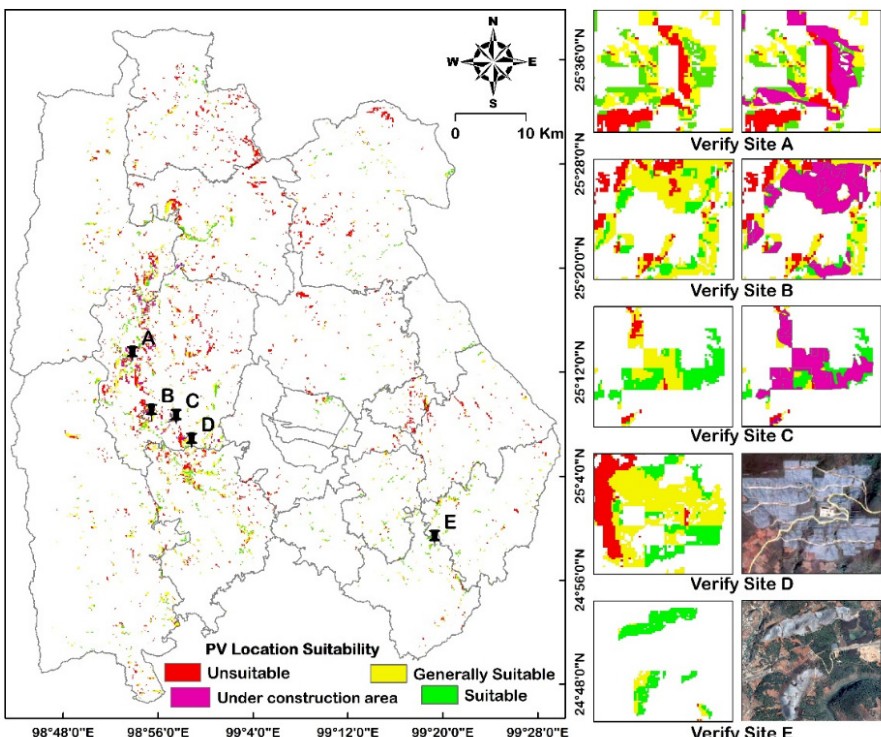

**Figure 7.** Verification and analysis of the suitability of photovoltaic site selection in Longyang District, Baoshan City.

The two completed PV power stations, D and E, located in Yangliu Township and Xinyi Township, are all constructed in suitable and generally suitable areas of the site selection results. The overlay analysis of the station area under construction covering an area of 709.2 hectares with this result indicates that 92.78% of the area is located in suitable and generally suitable areas. This study can accurately identify the potential area of photovoltaic power station construction. It can provide a reliable theoretical reference for the government and enterprises to decide on the location of a photovoltaic power station.

## 4. Conclusions

This study employed GIS combined with a FAHP to construct a photovoltaic site selection model, addressing the biases introduced by traditional methods and saving time while eliminating redundant work in the site selection process. It also overcame the difficulties associated with subjective factor weighting and model consistency testing in AHP and MCDM methods. Based on the Third National Land Use Survey data, five land-use types, including vacant land, bare land, shrubland, other forestland, and other grassland, were selected for site analysis. During this phase, unsuitable land types such as agricultural land and construction land were filtered out, reducing land acquisition costs for businesses. Considering the relatively high frequency of geological disasters in mountainous areas, this study included geological disasters as an influencing factor, which is relatively rare in previous research. The site selection method was validated using existing photovoltaic power station areas based on the site selection results. The study found that this site selection method has a high level of accuracy, providing a relatively accurate reference for government and businesses in site selection.

Spatial analysis reveals that within the study area, 45.77% of the land is unsuitable for the construction of photovoltaic power stations. These areas are primarily located to the north of Longyang District, characterized by higher elevations, steeper slopes, and closer proximity to geological hazard zones, making them unsuitable for development. On the other hand, 54.23% of the suitable areas are mostly situated in the southwestern part of Longyang District. Among them, Yangliu Bai and Yi Autonomous Township and Pupiao

Town have larger suitable areas, accounting for 25.3% and 16.1% of the total suitable area, respectively. In this region, the elevations and slopes are moderate, there is a considerable distance from geological hazard-prone areas, and the proximity to substations is closer, resulting in higher suitability for photovoltaic development.

Due to various constraints imposed by the local government and the unique topography of the research area, enterprises prioritize cost expenditures during photovoltaic construction, often overlooking the protection of local water sources and the environment. They may also neglect the potential impacts of unavoidable disasters after the station is completed. In this study, both the constraining factors in the early stages of photovoltaic station construction and the influencing factors after construction are considered. The assignment of weights to each factor is particularly crucial in the research process. Given that 80% of Yunnan Province is mountainous, with a high frequency of landslides, mudslides, and other disasters, the factor "Distance to geological hazards" obtained the highest weight in this study, with a weight of 0.163. "Distance to ecological protection zones" had a weight of 0.128, reflecting the local emphasis on ecological respect and protection. Slope aspect and elevation in the terrain received weights of 0.132 and 0.104, respectively. Compared to other image factors, the weights for annual average land surface temperature and annual average sunshine hours were slightly lower at 0.023 and 0.043, respectively. This study provides a potential and feasible solution for the complexity of site selection conditions faced by relevant power departments in Yunnan Province during the decision-making process for photovoltaic site selection. It presents an opportunity for Yunnan Province to transition to new energy sources. Having scientifically validated land resources can enable investors and decision makers to build more environmentally friendly and sustainable solar photovoltaic power stations.

Since the solar energy potential of the study area is between 1650 KWh/m$^2$/year and 1700 KWh/m$^2$/year, which is greater than the ideal value of global horizontal irradiance GHI [39,40], the global horizontal solar irradiance was not evaluated as a standard in this study. Considering soil resistivity and corrosive characteristics would make the research more comprehensive. However, due to a lack of relevant data, the above factors were not considered in this study. This study focuses on mountainous photovoltaic site selection, aiming to enable the government to familiarize itself with the areas within its jurisdiction that are suitable for the construction of photovoltaic power stations, and provide regional reference for investors of related enterprises in the process of selecting photovoltaic power station locations in mountainous areas. Considering policy restrictions and cost control, this study selected five types of land use with relatively low costs. In the future, with the support of policies and the improvement of industry standards, the inclusion of various land-use types such as parks and forests will make the land-use types in the photovoltaic site selection model more diverse.

**Author Contributions:** Methodology, Y.L.; software, Z.F.; validation, Y.L. and Z.F.; formal analysis, Y.L.; resources, Y.L.; data curation, Z.F.; writing—original draft preparation, Y.L., Z.F. and J.Z. All authors have read and agreed to the published version of the manuscript.

**Funding:** This research was funded by (1) the Multi-government International Science and Technology Innovation Cooperation Key Project of the National Key Research and Development Program of China for the "Environmental monitoring and assessment of land use/land cover change impact on ecological security using geospatial technologies" (2018YFE0184300).

**Data Availability Statement:** Data is contained within the article.

**Conflicts of Interest:** The authors declare no conflict of interest.

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
