# Peer review of "Location of Mountain Photovoltaic Power Station Based on Fuzzy Analytic Hierarchy Process—Taking Longyang District, Baoshan City, Yunnan Province as an Example"

_sustainability, doi:10.3390/su152416955_

Round 1
Reviewer 1 Report
Comments and Suggestions for Authors
The authors carried out interesting research considering the location of mountain photovoltaic power stations. The applied methodology is based on the Fuzzy Analytic Hierarchy Process that can be considered as an appropriate method for solving such multi-criteria problems. I would suggest certain improvements to the paper before it could be suitable for publication.
1. There is a suggestion regarding citing the references in the text. As a rule, the references during the text should be cited just by author's surname, instead of both surname and name. For example, it should not be:
„Majumdar D et al. employed MCDM to evaluate the suitability of photovoltaic development land in Arizona based on terrain, location, and solar energy resources“
2. Further, the citations should be placed the first time they are cited. For example, the citation is missing here:
„Bahaa E and others considered the constraints of topography, environment, meteorology, and climate when evaluating the suitability of different regions in Egypt for the construction of photovoltaic power stations.“
3. The statements in Table 3 need more discussion. For example, it should be further explained why a higher “Distance from the highway” brings to better suitability level of photovoltaic site selection.
4. In the calculations of the factor weights in the fuzzy analytic hierarchy process, it is not clear how the fuzzy judgment matrix of all indicator levels is formed. Who gave the answers, are there any experts involved?
5. The Section “3.1. Data processing results” is not clear enough. There is a statement “Each factor was then subjected to overlay analysis according to the weights determined using the Fuzzy Analytic Hierarchy Process (Fuzzy AHP), and the results are presented in Figure 3”. However, Figure 3 does not give sufficient information. It would be welcome to provide some more numerical results.
6. Verification of the suitability of Longyang District photovoltaic site selection is performed by comparing the results with the locations of constructed or under-construction photovoltaic power stations in the research area. However, is it possible to apply also some other MCDM model, in order to confirm the obtained results?
Author Response
We sincerely appreciate the valuable feedback from the editors and all the reviewers. Their comments and suggestions have provided excellent guidance for improving the quality of our manuscript. Below are our responses and explanations for the revisions made in accordance with the comments from the reviewers.
- There is a suggestion regarding citing the references in the text. As a rule, the references during the text should be cited just by author's surname, instead of both surname and name. For example, it should not be:„Majumdar D et al. employed MCDM to evaluate the suitability of photovoltaic development land in Arizona based on terrain, location, and solar energy resources“
Review Response: I have made revisions based on the experts' suggestions.
Modification Explanation: I have diligently addressed similar issues throughout the entire article. Thank you for bringing them to my attention. In our future writing, we will pay closer attention to these writing details to avoid similar errors.
- Further, the citations should be placed the first time they are cited. For example, the citation is missing here:Bahaa E and others considered the constraints of topography, environment, meteorology, and climate when evaluating the suitability of different regions in Egypt for the construction of photovoltaic power stations.“
Review Response: I have made revisions based on the experts' suggestions.
Modification Explanation: Corrections have been implemented, and we will carefully cross-check during the subsequent writing process.
- The statements in Table 3 need more discussion. For example, it should be further explained why a higher “Distance from the highway” brings to better suitability level of photovoltaic site selection.
Review Response: I have made revisions based on the experts' suggestions.
Modification Explanation: We believe that providing further descriptions of relevant constraint data enhances the overall readability of the article. Therefore, we have provided additional explanations for the factors related to suitability in Table 3. Please refer to sections 2.3.1 to 2.3.4 for detailed explanations.
- In the calculations of the factor weights in the fuzzy analytic hierarchy process, it is not clear how the fuzzy judgment matrix of all indicator levels is formed. Who gave the answers, are there any experts involved?
Review Response: During the FAHP modeling process, we consulted experts with backgrounds in engineering, environmental studies, and sociology.
Modification Explanation: To provide a clearer expression, we evaluated the importance of various factors involved in the photovoltaic site selection process and obtained the importance levels of each criterion. These assessments are presented in Table 5. Please refer to Table 5 for detailed information.
- The Section “3.1. Data processing results” is not clear enough. There is a statement “Each factor was then subjected to overlay analysis according to the weights determined using the Fuzzy Analytic Hierarchy Process (Fuzzy AHP), and the results are presented in Figure 3”. However, Figure 3 does not give sufficient information. It would be welcome to provide some more numerical results.
Review Response: I have made revisions based on the experts' suggestions.
Modification Explanation: We have improved Figure 3 to better align with the suitability levels as indicated in Table 3. This adjustment allows for a more effective representation. Please refer to Figure 3 for details.
- Verification of the suitability of Longyang District photovoltaic site selection is performed by comparing the results with the locations of constructed or under-construction photovoltaic power stations in the research area. However, is it possible to apply also some other MCDM model, in order to confirm the obtained results?
Review Response: We conducted experiments using the MCDM model, but the results were not as promising. During the MCDM model process, we observed that the subjective assignment of weights to various factors played a significant role, leading to less optimistic model performance. In contrast, the stability of experimental results was not as robust as that of the FAHP model.Given that both completed and ongoing photovoltaic power plants undergo extensive discussions involving government energy departments and industry experts during the initial site selection phase, taking into account social and natural conditions, the selection of photovoltaic land in Longyang District is based on these deliberations. Investors and decision-makers also conduct on-site inspections after the site is determined. In this experiment, the data derived from these real-world validations have higher credibility and authority. It better reflects the accuracy and practicality of the model used in this study. We believe that using this data for validation adds a higher level of scientific rigor to our research.

Reviewer 2 Report
Comments and Suggestions for Authors
The authors discuss an interesting topic connected to sustainability issues. In particular, the article focuses on the integration between GIS and FAHP methods for solar station locations. The case study is applied to a region in China. The paper describes all geographical data used and analyzed and their sources.
The main topic of the article is about the use of AHP methods for the selection site for a photovoltaic power station. The topic is interesting and relevant not only for the processing in GIS environment of data but also as a support method for a conscious choice for a solar station set. The paper define not only a method to be followed for similar application but define also the set of data to be collected for the site selection. Due to the issues concerning the sustainability of the photovoltaic power stations in natural context the parameter defined should consider also the environmental impact of these plants.
Conclusions are related to data and information discussed.
It could be interesting to better describe how data connected to all constraints are analyzed.
Value of the accuracy of site selection has to be discussed in depth.
Images and tables are adequate.
References could be improved.
Author Response
We sincerely appreciate the valuable feedback from the editors and all the reviewers. Their comments and suggestions have provided excellent guidance for improving the quality of our manuscript. Below are our responses and explanations for the revisions made in accordance with the comments from the reviewers.
Conclusions are related to data and information discussed.
Review Response: I have made revisions based on the experts' suggestions.
Modification Explanation: After careful examination by project team members, it was found that the conclusion section indeed lacked a discussion of the research data and information. We have supplemented the conclusion section to ensure a close correlation between the discussion content and the data.
It could be interesting to better describe how data connected to all constraints are analyzed.
Review Response: I have made revisions based on the experts' suggestions.
Modification Explanation: We believe that providing further descriptions of relevant constraint data enhances the overall readability of the article. Therefore, we have provided additional explanations for the factors related to suitability in Table 3. Please refer to sections 2.3.1 to 2.3.4 for detailed explanations.
Value of the accuracy of site selection has to be discussed in depth.
Review Response: I have made revisions in accordance with the experts' suggestions.
Modification Explanation: We have reviewed relevant literature to explore the topic. In the section "4.2 Validation of Photovoltaic Site Suitability in Longyang District," we have incorporated a discussion on the significance of site selection, highlighting the importance of the chosen locations. This addition enriches the content of the article and sets the stage for the validation of our study.
Images and tables are adequate.
Review Response: Due to the large size of the images, we were unable to insert them directly into the manuscript. We have uploaded high-resolution images to the attachment for your clarity.
References could be improved.
Review Response: I have made revisions based on the experts' suggestions.
Modification Explanation: We have thoroughly reviewed and standardized the format of all the references. Specifically, we have made modifications and additions to the references section to make it more comprehensive and accurate. Additionally, we have corrected the citation format.

Round 2
Reviewer 1 Report
Comments and Suggestions for Authors
The authors improved the manuscript according to the suggestions of the reviewer.